# Emulating and Evaluating Virtual Remote Laboratories for Cybersecurity [note 1]

**DOI:** 10.3390/s20113011

**Published:** 2020-05-26

**Authors:** Antonio Robles-Gómez, Llanos Tobarra, Rafael Pastor-Vargas, Roberto Hernández, Jesús Cano

**Affiliations:** 1Department of Control and Communication Systems, Computer Science Engineering Faculty, Spanish National University for Distance Education (UNED), 28040 Madrid, Spain; llanos@scc.uned.es (L.T.); rpastor@scc.uned.es (R.P.-V.); roberto@scc.uned.es (R.H.); jcano@scc.uned.es (J.C.); 2Faculty of Law, San Pablo CEU University, 28003 Madrid, Spain

**Keywords:** emulated virtual scenarios, vulnerability analysis, practical skills, online resources, UTAUT/TAM factors, SEM validation

## Abstract

Our society is nowadays evolving towards a digital era, due to the extensive use of computer technologies and their interconnection mechanisms, i.e., social networks, Internet resources, IoT services, etc. This way, new threats and vulnerabilities appear. Therefore, there is an urgent necessity of training students in the topic of cybersecurity, in which practical skills have to be acquired. In distance education, the inclusion of on-line resources for hands-on activities in its curricula is a key step in meeting that need. This work presents several contributions. First, the fundamentals of a virtual remote laboratory hosted in the cloud are detailed. This laboratory is a step forward since the laboratory combines both virtualization and cloud paradigms to dynamically create emulated environments. Second, this laboratory has also been integrated into the practical curricula of a cybersecurity subject, as an additional on-line resource. Third, the students’ traceability, in terms of their interactions with the laboratory, is also analyzed. Psychological TAM/UTAUT factors (perceived usefulness, estimated effort, social influence, attitude, ease of access) that may affect the intention of using the laboratory are analyzed. Fourth, the degree of satisfaction is analyzed with a great impact, since the mean values of these factors are most of them higher than 4 points out of 5. In addition to this, the students’ acceptance of the presented technology is exhaustively studied. Two structural equation models have been hypothesized and validated. Finally, the acceptance of the technology can be concluded as very good in order to be used in other Engineering contexts. In this sense, the calculated statistical values for the improved proposed model are within the expected ranges of reliability (X2 = 0.6, X2/DF = 0.3, GFI = 0.985, CIF = 0.985, RMSEA = 0) by considering the literature.

## 1. Introduction

The interest in developing and employing new technologies has increased in recent years, even more, with the high accessibility to Internet resources and services, seen as Anything as a Service (XaaS) [1]. The daily life of many people is evolving to use digital resources from remote locations and considering the ubiquity of users. There is a need for training professionals in many areas of the society, such as e-Health [2], marketing for sales [3], or distance teaching/learning [4], among other areas.

In the field of education, the learning–teaching process can be supported by the laboratory as a service approach [5]. These kinds of learning/teaching resources become even more relevant with distance education methodologies, as the National University of Distance Education in Spain is (named in Spanish as UNED), since students do not interact face to face in the classroom. Practical activities at a distance are essential to satisfy the corresponding competences and learning’s objectives of this methodology.

The primary topic of this work is the distance education of cybersecurity. In this case, the learning process of students has additional challenges, such as keeping the interest of students and minimizing drop-outs. It is also very relevant to have a fluent virtual communication among students and with lecturers. This fact eases their guidance to achieve the competences and abilities defined in the curricula, as well as using the proposed resources into the online learning platforms. Our future Engineers, specialized in cybersecurity, must be able to address the possible technological threats on the Internet, not only in a theoretical way but also in a practical way [6,7,8]. In addition to this, a set of useful tools for increasing the motivation and interest in cybersecurity can be found in [9,10,11]. They employ games and competitions based on the capture-the-flag (CTF) approach to keep students engaged with the learning process and improving qualifications at the same time.

In this sense, this work proposes and evaluates exhaustively a new Virtual Remote Laboratory (named ViRe-Lab) for the cybersecurity topic, hosted in the cloud employing virtualization technologies. The proposed laboratory, in which a set of practical activities can be defined, is built over the EVE-NG technology (Emulated Virtual Environment – Next Generation [12]). Lecturers are able to define emulated and very realistic laboratories on remote with minimum configuration requirements [13]. The results obtained from the assessment of the laboratory are divided in several parts, in terms of interactions with the laboratory, satisfaction of use it, and the students’ acceptance of the technology.

The proposed laboratory will be used in the “Cibersecurity in Information Systems” (CIS) subject (*in Spanish, “Seguridad en los Sistemas de Información”*). Employing the online platform of UNED, named aLF (active Learning Framework) [14], the contact between the teaching team and the students can be continuous, as well as the interrelation between the students themselves (asynchronous forums). The UNED University also manages a significant amount of centers, and extension centers in small towns which are distributed throughout the Spanish geography and other countries in Europe, Africa, Asia, and America. They are more than 300 centers, where students can optionally attend face-to-face classes, so the UNED methodology is blended, but only for degree subjects. However, the CIS subject belongs to a post-degree in Computer Science Engineering. The proposed practical activities are carried out entirely on distance.

It is also necessary (in distance education) that the resources offered to students are accessible and with a high quality. Therefore, the proposed remote laboratories must support a quality of service without expecting fails or delays and problems of access. In this sense, the acceptance of this technology must also be considered. It is necessary to have some acceptance and use theory for technology, such as UTAUT [15]. This is a very suitable model to analyze the intention to use the technology, and analyzing their benefits and drawbacks, but also taking into account the students’ behaviors. According to this, there are a set of elements to be elicited to obtain useful psychological factors. Another useful model to create and validate the user’s intentions to employ a new technology is TAM (Technology Acceptance Model) [16], as well as many variations and extensions [17,18]. In recent years, some research works have proposed mixed and integrated UTAUT/TAM models, such as [4,19], considering the advantages of both of them.

Therefore, the concrete contributions of this work are: (1) the creation of a virtual remote laboratory, ViRe-Lab, which allows lecturers to create emulated realistic scenarios in the cloud without configuration requirements for the students;(2) the integration of the laboratory within the curricula of a post-degree subject about cybersecurity; (3) the analysis of the students’ traceability to observe their interactions with the technology; (4) studying the grade of satisfaction considering a set of psychological factors, when students interact with the laboratory; and (5) analyzing the acceptance of the laboratory to be used it for other Engineering contexts. Two structural equation models have been hypothesized and validated for our proposed laboratory. These equation models follow the guidelines provided by the UTAUT/TAM model. A set of statistical values have also been calculated to measure if the second proposed model is valid and within the expected ranges of reliability, according to the examined literature.

This manuscript is structured as follows. Section 2 presents the state-of-the-art and a summary of previous works on remote virtual laboratories, some of them oriented to cybersecurity. After that, Section 3 describes the educational context, and presents our starting hypotheses of the study presented in this work. Section 4 presents the implemented remote virtual laboratory for cybersecurity, and a practical activity integrated in a cybersecurity subject. The obtained results from the experience with students based on the proposed laboratory are presented in Section 5. Finally, conclusions and further work are given in Section 6.

## 2. State of the Art

When managing an Infrastructure as a Service (IaaS), it is essential to perform an efficient use of the available resources attending the previously established service level agreements and operation level agreements (SLAs/OLAs). In [20], the employed infrastructure to provide services to costumers is optimized from both the point of view of resource provisioning in the cloud, as well as the satisfaction of costumers. In practice, authors propose a hiring resource model for providers by considering several parameters, such as SLAs, resource allocation, satisfaction, and costs, in order to maximize revenues. Simulation results obtained with CloudSim highlight the revenue optimization and customer satisfaction.

Within the Internet-of-Things (IoT) topic, some proposals have also taken into account the aspect of security in different IoT layers: sensors and smart devices, distributed edge computation, and the network communication in the cloud. An example is [21], in which a security IoT framework for organizations is proposed considering the three mentioned layers. Cybersecurity is also very relevant at the level of protocols and algorithms. For instance, in [22], a secure authentication protocol for Radio Frequency Identification (RFID) systems is proposed and analyzed. Another example of managing technologies in an efficient way can be found in [23]. In here, authors proposed the SensGrid simulation toolkit for experimentation in the field of Wireless Sensor Networks (WSNs) with Grid computation for the resource allocation.

Focusing on the educative laboratories, higher institutions and universities nowadays have the need of providing students with physical equipment from a remote location for training in practical skills, and gaining professional abilities within the field of Engineering, as well as emulated scenarios. Another approach would be the use of immersive virtual reality [24]. When a distance methodology is used, as in the case of UNED, it is very relevant to integrate virtual remote laboratories, seen as additional resources, into the curricula of Engineering subjects, such as the cybersecurity topic. Remote equipment or laboratories can be seen as several hardware and software elements, which work together to provide a set of functionalities. They can be seen as individual resources or a set of services. The laboratory itself is located in a different physical location than laboratory users. These users can use it for some reason, such as educational purposes. A variety of remote labs have been defined and created for several fields throughout the last years. Some proposals are [25,26,27]. An Internet connection is only needed to access the remote laboratory.

Distance laboratories (remote or virtual) are used in different educational areas. Related to renewable energies in the context of distance education, low-cost remote laboratories have been developed in recent years [28]. The concept of deconstruction of laboratories was introduced, in which a remote laboratory is made up of many services. From the client perspective, all services or a subset of them can be used to adapt the learning process to students. This approach is known as LaaS (Laboratory as a Service) [5]. The work was conducted in the context of the MUREE project [29,30]. Each component is used as a web service, which can be hardware or software. It is possible that the hardware can only be used in an active way by one single user at a time, hence it is necessary to implement control protocols for these hardware elements.

Remote laboratories can be managed by resources to keep the quality of online teaching in Engineering courses [31]. Another relevant work is [32], where the evolution of a set of learning laboratories is presented for the learning process of distributed network services and cybersecurity, to improve the students’ grades and minimizing drop outs into distance courses [33]. Another recent work [4] pays attention to the development and assessment of remote laboratories for renewable energies. It validates the successful integration of remote laboratories by defining a structural equation model. Authors conclude that these kinds of laboratories are useful to improve the quality of virtual courses at a distance.

As a complement of a remote laboratory, the concept of virtual laboratory appears. This way, only software elements compose the laboratory. Virtual laboratories can be used for emulating physical laboratories or adding new features to remote laboratories through their software components, as an alternative or for reinforcing their use. [34] proposes a virtual laboratory to configure and evaluate network services automatically. Each student had to configure his/her network server with a set of services (DHCP, DNS …) and, later, the virtual lab was able to detail the configuration problems of these network services. In this sense, the virtualization paradigm [35,36,37] has additional benefits for the dynamic creation of virtual laboratories. In general terms, time and processing resources are usually consumed more effectively and efficiently by the technical staff when using this kind of technologies. Some efforts have already performed in the context of laboratory virtualization with a distance methodology at UNED [38].

As a further step, virtualization and cloud paradigms are merged here for dynamically providing students with useful emulated environments The current work for the creation of the remote virtual laboratory and its dynamic scenarios for practical activities is an exhaustive extension of [13]. This way students can acquire practical Engineering skills in a context in which no physical interactions occur among students. Students develop critical assets for their professional careers such as practical experience and autonomous work. Closely related to the creation of virtual laboratories to help teachers in schools and other institutions was the Go-Lab project [39], in which UNED was also involved. These types of virtualized laboratories contribute to the sustainability of university education on distance [40]. This fact has much impact at UNED, where the number of users in virtual courses could be massive.

As an additional step, we analyze the students’ learning experiences. Several SEMs (Structural Equation Model), have also been defined and validated statistically by using the students’ opinions. To define our SEMs, we have based our findings out in the UTAUT methodology [15]. There are other many kinds of models, but the UTAUT model is nowadays one of the most popular, as already justified in previous works for the field of Engineering, one of the most recent ones is [4]. Another interesting research work using UTAUT models for the field of health care technology can be found in [19].

## 3. Methodology

### 3.1. Instruments and Demography

The “Cybersecurity in Information Systems” (CIS) subject is part of the Master (MSc) in “Computer Science Engineering” at the UNED School of Computer Science Engineering. The Master is made up of a set of compulsory and optional subjects, some of them with six ECTS credits and others with four ECTS credits. The subject considered in this paper is mandatory, consists of four ECTS credits and is studied in the first semester of the first academic year.

The methodology employed is detailed in this section. First, the educational context for this work is presented, by including the competences and learning results of the subject used here. The provided contents will be linked with the emulated virtual laboratory presented in Section 4. In particular, the case of study will exhaustively be described.

In addition to this, a set of initial hypotheses are defined in this section, in order to analyze the acceptance of the presented technology in Section 5. These hypotheses consider a set of psychological factors, which are the core of a TAM/UTAUT model, and they have already been used in our previous findings for other educational contexts [41]. These hypotheses will be refined as a second step.

As for the demographic data twenty students finished the practical activities with the ViRe-Lab and answered the opinion survey, more or less the 70% of the enrolled students in the CIS subject. This activity was mandatory. Among the students, 17 out of 20 were male and three out of 20 were female. The population was relatively young, since only the 20% of the students were 40 years old or more. The job of most of the students were related to computer science, a total of the 80% of them. Table 1 also shows the corresponding demographic data more in detail.

### 3.2. Educational Contextualization

Attending the teaching framework of the CIS subject, although Computer Science Engineering does not have officially recognized professional attributions yet, this post-degree was designed by following the specifications of the Conference of Deans and Directors of Computer Science Engineering (*In Spanish*, Conferencia de Decanos y Directores de Ingeniería Informática, CODDII) in Spain. To pass the whole post-degree, it is necessary to have passed a total of 90 ECTS credits, and a set of optional training complements if needed, that the student needed to fulfill the competences required for accessing to the post-degree.

The specific competences of the subject are the following:Capacity for strategic planning, development, management, coordination, and technical and economic management in the fields of Computer Science Engineering, among others, with: systems, applications, services, networks, or computer facilities and software development centers or factories, respecting the adequate fulfillment of quality and environmental criteria and in multidisciplinary work environments.Ability to design, develop, manage and evaluate mechanisms for certification and guarantee of cybersecurity in the treatment and access to information in a local or distributed processing system.

The Learning Results (LR) that cover these competences are the following:LR1. Understand advanced concepts of cybersecurity in the treatment and access of data.LR2. Know advanced mechanisms of certification and guarantee of cybersecurity.LR3. Understand challenges and cybersecurity solutions of within the context of the Internet.LR4. Design, development, and management of cybersecurity mechanisms.

Therefore, the syllabus of the subject will cover the following contents, each of them associated to one specific LR:*Advanced design of a cybersecurity program*. In this unit, students are introduced to the cybersecurity programs within the Information Security Management Systems (ISMS). In particular, the fundamental concepts of ISO–27001 standard is introduced, which allow us to develop a successful safety program. This standard is also compared to other existing ones in the literature. The primary objective of any cybersecurity program is to mitigate the risks that may occur within the organization by implementing this program. The mitigation concept does not consist of eliminating them, but reducing them to an acceptable level, taking into account various economic, social, structural parameters, etc. To ensure that the cybersecurity elements deployed in an organization control the risks, it will be necessary to anticipate the incidents that may arise, through the various techniques of risk analysis.*Advanced cybersecurity models in information systems*. Modern cybersecurity products are designed to strike a balance between the needs of businesses on the Internet. Within this unit, we will offer an initial vision of the latest technologies that allow future cybersecurity professionals to take into account the most modern threats in the design of their cybersecurity programs.*Management of cybersecurity operations*. We will focus on the metrics and Key Performance Indicators (KPIs) that we can use to monitor compliance. These measures, being some qualitative and others quantitative, allow us to determine how close we are to fulfill a cybersecurity program in the organization in terms of time, costs, percentages, and so on. These metrics and KPIs refer to the performance of security policies (incidents, vulnerabilities, updating, configurations, changes …), which are not related to the psychological factors studied and analyzed in the paper.A cybersecurity program is not a static process, but will undergo several changes over time, offering improvements. We will also deal with change management and its documentation in this module. We also describe the creation of controls that verify when the cybersecurity program is fulfilled or not within our ISMS. Methodologies that allow us to verify compliance with the cybersecurity program as the guide that the ISO–27001 standard offers us through these controls are detailed.*Monitoring, recovery, and response to vulnerabilities*. Within this unit, we will concentrate on the monitoring of the cybersecurity of our systems, as well as its audit. This procedure should indicate to the administrators of the system that operates in an expected manner, where the faults are located, as well as the load supported by the system. It also allows to expose suspicious activities, audit trails of system use, as well as forensic evidence that is useful in the subsequent diagnosis of attacks. On the other hand, the unit focus on the practices necessary to provide the services offered by the organization reliability and recovery. It also address issues related to advanced backup schemes in a recovery scenario, as well as high availability of the services of the organization to ensure its continuity of operation.

### 3.3. Initial Hypotheses

This work also aims to analyze the theoretical factors that influence students in their learning process, when interacting with the proposed laboratory. To perform this analysis, two SEMs will be defined and analyzed from students’ acceptance with the emulated laboratory. This type of model is a general multivariate analysis technique, which hypothesizes and tests the causal relationships between the factors with a system of linear equations [42,43].

A SEM model is visually represented as a directed graph where nodes are factors and arrows the influence among them. The factors defined in the initial model will be based on an integrated UTAUT model defined in [19], from both the original UTAUT and TAM factors. The concrete factors to be analyzed in this work are: perceived usefulness, estimated effort, attitude, social influence, ease of access, and intention of use. Our initial hypotheses for our SEM models are Hypothesis H1, H2, H3, H4, and H5.

**Hypothesis 1** **(H1).**
*The perceived usefulness using the laboratory will positively influence the attitude of students to perform the practical activities.*


**Hypothesis 2** **(H2).**
*The estimated effort using the laboratory will positively influence the attitude of students to perform the practical activities.*


**Hypothesis 3** **(H3).**
*The attitude of students to use the laboratory will positively influence the intention of use it in the future.*


**Hypothesis 4** **(H4).**
*The social influence among users of the laboratory will positively influence the intention of use it in the future.*


**Hypothesis 5** **(H5).**
*The ease of access to the laboratory will positively influence the intention of use it in the future.*


## 4. Emulating Virtual Remote Laboratories in the Cloud

Our emulated virtual remote laboratory, ViRe-Lab, has been implemented taking as a basis the Emulated Virtual Environment - Next Generation (EVE-NG) [12] technology. The principal characteristics of the EVE-NG core are the efficient management of the EVE-NG core, the dynamic and graphic creation of the network topology, a straightforward definition of laboratories and configuration files in the cloud, local memory optimization, responsive client-interfaces, the possibility of integrating real and virtual devices, and the simultaneous laboratory instantiating, among others. The concepts of virtualization and cloud provision fit this approach.

### 4.1. Creation of the EVE-NG Virtual Environment

In order to create and deploy emulated virtual scenarios with EVE-NG, which are hosted in the cloud, it is necessary to follow a set of phases: (1) Obtaining the virtualization software and remote connection tools; (2) Creation of the emulated virtual environment; (3) Updating access permissions; (4) Adding the required virtual resources to the emulated environment; (5) Starting the EVE-NG core and deploying an automatic setup script; and (6) Defining practical scenarios for the corresponding subjects. Details about these points are the following:Obtaining the virtualization software and remote connection tools. The virtualization software can be based on Virtual Box, VMware, Docker containers, or a similar virtualization technology. Additionally, remote connection tools will be useful to graphically access the virtualization resources located in the EVE-NG environment, such as Telnet, VNS, among others. These tools have to be installed into the network infrastructure where hosted the virtualized EVE-NG environment.Creation of the emulated virtual environment with EVE-NG. It is necessary to download and install the virtualized EVE-NG core within the network infrastructure as the principal virtual resource, which controls all resources. For instance, virtualization could be based on a VMWare ESXi server.Updating access and configuration permissions. All the access permissions to the network interfaces have to be specified according to the pre-established access policy. The same happens with the interface configuration files. These permissions can be updated later.Adding the required virtual resources to the EVE-NG environment. The desired images and VMs for emulating the different devices of the network scenarios have to be incorporated to the virtualization server. This fact depends on the concrete scenarios defined for the practical activities of the subjects. The set of available resources can dynamically be updated later.Starting the EVE-NG core and deploying an automatic setup script. Once the emulated EVE-NG environment is created, this platform has to be restarted with a concrete configuration, as well as a set of scripting lines to automatically re-init system correctly when it fails, turns off, etc. With this technology, real and emulated devices can be mixed at the same time.Defining practical scenarios for the corresponding subjects. A set of sample scenarios have to be defined and tested prior to deployment for production purposes. As an example, Figure 1 represents a part of a virtual laboratory with a network facility defined in ViRe-Lab for audit purposes. In particular, there is an access point for the auditor, another machine with a set of audit tools (available in a Kali Linux distribution), and a local router connected to the access point and Kali. Prior to enter the laboratory, but not shown here, the user finds information about the own working laboratory and some guidelines about the practical activity.

### 4.2. The Purpose and Access of the Laboratory

The principal purpose of the proposed virtual remote laboratory, ViRe-Lab, is to develop practical activities in the context of Engineering, like the cybersecurity topic. The registration screen is shown in Figure 2. Our virtual remote laboratory, ViRe-Lab, allows the user’s interaction with the devices of a network facility defined by the lecturer. A student may use all nodes and routes as real equipment, depending on the permissions provided by lecturers during the preparation of the practical activity.

A session and booking layer is incorporated and integrated into the core of EVE-NG, to control the user access to ViRe-Lab, in terms of available resources. The Algorithm 1 shows the procedure to control the access to ViRe-Lab that includes a session calendar. Each student can book a free session by choosing one or several days and hours from the calendar. A specific account is needed to enter the platform. ViRe-Lab incorporates a scheduler to allow several students in the laboratory. However, only a user at the same time has access with an active role.
**Algorithm 1** Access Control and Booking Algorithm for ViRe-Lab. **procedure**
Laboratory Access and Booking (*UserData*, *booking*)  *Calendar* ← Data structure with free and busy slots of time  **if** ((*booking*)&&(*AvailableSlot*(*Calendar*)) **then**   *BookingSession(UserData, Calendar)*   message(“Booked a selected and free session in the calendar.”)   **return** True  **else**   **if** (!*BookedSlot*(*UserData*, *Calendar*)) **then**    *message(“There is not a booked slot session for the user at the moment. Yo can do it.”)*    **return** False   **else**    *Resources* ← List of available virtual resources    **if** (!*AvailableResources*(*Resources*)) **then**     *message(“There is not available resources at the moment. Try it later, please.”)*     **return** False    **else**     *UserResources* ← List of booked virtual resources for this user     *CheckingConfigurations(UserData, UserResources)*     *RunVirtualLab(UserData, UserResources)*     *message(“Setting up the laboratory for user access.”)*     **return** True

As observed in Algorithm 1, in case the student desires to access directly to the laboratory, the system would check if he/she had a booked session. Also, ViRe-Lab examines if there are available virtualization resources for the current session. If so, it assigns the session a set of virtual resources, checks the correctness of all configuration files, and deploys a virtual network defined by software. Finally, the student enters the system for performing practical activity. Additionally, he/she has the possibility of checking his/her pending and completed sessions. Each session lasts up to 55 min.

### 4.3. Practical Activity

The presented case of study is related to the practical activity proposed by the CIS subject to be developed with ViRe-Lab since this is integrated into the on-line course as an additional resource. It is composed of a set of objectives, which correspond to the competences, learning results, and contents of the subject. These objectives are detailed next in a descriptive and incremental way, depending on the skills achieved by the student during the course:*Analyzing the context of the company in a non-intrusive way*. The search for non-intrusive information focuses on obtaining as much information as possible from the company system with no knowledge of the organization’s internal structure. Students must avoid techniques that are considered intrusive (to be used in the following steps). Therefore, a student will be take the role of a penetration tester (pentester) recruited by a fictional company, named *Disaster Corporation*. As initial tasks, he/she can visit the company portal and obtain information about the network facility of the company The student can access the system through an unique access point in the cloud, as shown in Figure 3. He/she will not know any details about facility of the company.*Discovering the network structure of the company in both a non-intrusive and intrusive ways*. Some initial tools can be employed by students to discover information about the network system: (1) the *nslookup* tool, in order to make queries to the DNS servers, so obtaining information about a possible target; and (2) the *ping* tool, in order to find out accessible and non-accessible IP addresses. It is worth mentioning the *traceroute* tool that allows detecting the route followed on the Internet to reach an IP address [44]. In addition to this. The first intrusive step will be to locate those computers that are turned on and connected to the network (devices, ports, and services) with the *nmap* tool [45]. We can find out which network we are connected to by using the *ifconfig* (Linux) or *ipconfig* (Windows) command. As observed in Figure 4, the company network of the practical activity is composed of five devices (from 192.168.56.1 to 192.168.56.5). The network IP (192.168.56.0) was found out with both the *nslookup* and *ping* tools.*Finding out weak access points (open ports) in the network structure of the company*. After knowing which devices are active into the network, it is interesting to find out possible access points to those computers. Therefore, the student must pay attention of the existing open ports. In the previous step, the devices of the network were discovered. It is now necessary to scan one-by-one with the *nmap* tool, instead of scanning the whole network, in order to discover the active ports of each of them. This will allow us to work faster, since our all information about our own devices is already in our own.*Discovering active services using the detected the access points (or devices)*. The next step is to determine which services are running associated with each port. At this point, if we found a machine with all ports closed, we would ignore it, prior to do a more exhaustive scan. In particular, the open ports and associated services belonging to a set of network devices are shown in Figure 5. From this analysis, we obtain additional information to verify existing vulnerabilities into the network facility of the company.*Achieving all the possible information about possible targets*. It is also needed to obtain more detailed information about each device of the company, which is a possible attack target, such as operating system, users, computer names and any information we may locate. It is also possible to perform this enumeration with the *nmap* tool. This command option usually takes time and generates large amounts of traffic. It also produces extensive output information. As an example, all information about the *ssh* service (open port 22), the operating system located in the host, keys, version of protocol, etc., is shown in Figure 6.*Identifying and reporting vulnerabilities*. From the previous findings, the student will have to identify the most relevant vulnerabilities found in the network system, and to make a report about these and how to solve them. Therefore, the cybersecurity policy of the company will be updated.*Exploiting vulnerabilities*. From the information collected in the report, the student must check the redefined cybersecurity policy with the role of a pentester. This will allow us to detect which system vulnerabilities can be exploited and to implement some known exploit to see if it is possible to find an attack in the system.

## 5. Results

### 5.1. Experiment Setup

The evaluation process consists of several steps: (1) Studying the interaction of students with the ViRe-Lab, in terms of access, booking sessions, and geographical location; (2) Analyzing the students’ satisfaction from a set of psychological factors [19]: perceived usefulness, estimated effort, social influence, ease of access, attitude) to determine the intention of using the technology; and (3) analyzing the students’ acceptance of the presented technology to be considered it for other Engineering contexts. Two structural equation models are hypothesized and validated. A set of statistical values have to be calculated to confirm the goodness of the second proposed model in terms of reliability [4,15].

The participants had the possibility of filling out an opinion survey to measure their satisfaction and acceptance, when performing the practical activities in the ViRe-Lab for the CIS subject. Each factor was made up of a set of related questions, based on previous studies [4,19]. The selected factors are perceived usefulness, estimated effort, attitude, social influence, ease of access, and intention of use. Each question is composed of a five-point liker-type scale, from (5) “strongly agree” to (1) “strongly disagree”.

### 5.2. Students’ Interactions

The amount of sessions employed for each student in the ViRe-Lab is represented in Figure 7. Horizontal axis corresponds to the identificator of the student, and vertical axis to the amount of sessions. The lecturers’ identificator (1 and 2) have been removed from the plot for clarity, since they were testing sessions. Lecturers’ identificators 6 and 14 have also been removed without any session, since they correspond to the identificators 7 and 15. These students registered twice into the ViRe-Lab laboratory. The proposed practical activity might be completed in a range of one to three sessions, depending on the skills and knowledge of students, except some concrete situations. This means that perceived usefulness and ease of access to the ViRe-Lab is good.

In addition to this, Figure 8 shows the amount of bookings per student into the ViRe-Lab, which have been divided by the concrete status of the session: *Execution*, *Ended*, or *Not Use*. Since each student session lasts up to 55 minutes, sometimes, there is not enough time to finish a planned part of the practical activity for that session. We can observe it is more usual to finish the session correctly. A small number of the sessions were not used, this means that the students logged in into the system, although he/she did not start the activity at that time.

The results presented in Figure 8 enforce the conclusions already obtained in Figure 7. It also clarifies that some of the sessions are employed by students, in order to make confidence with the system, prior to start the activity. Some minor differences between access sessions and reservation sessions can be found in the figures. For example, the user with identificator 7 reserved a total of 15 sessions, but the number of executed sessions was seven. This is because of the fact a set of booking can be done in a unique access session and/or it is possible to miss booking sessions. To sum up, the general tendency of both plots are similar.

On the other hand, Figure 9 shows the geographical distributions of students. In particular, this plot represents the students located in Iberian Peninsula. Although it is not shown in here, some students from the Canary Islands also participated in the practical activity, as well as from other countries.

Finally, the evolution of the booking distribution during the period of more activity with respect to the practical activity, from the 29th of December to the 12th of January, into the ViRe-Lab laboratory is represented in Figure 10. The amount of sessions around Christmas holidays is lower than after that period. There is a noticeable peak in the plot for the date of the 9th of January. The deadline of the practical activity was at the end of January, and they had to perform an exhaustive report before that date.

### 5.3. Students’ Satisfaction

As an exploratory phase, Figure 11 represents the mean values for each factor in a radar chart, whereas both mean and standard deviation are represented in Table 2. It can be observed that the usefulness and effort to use the laboratory are 4.30 and 4.55 out of five points, respectively. The attitude when using the laboratory is even better, a total of 4.61 points. The intention of use of the ViRe-Lab value for futures experiences is quite good, 4.40 out of 5.

However, the social influence during the experience with the laboratory is worse with a value of 4.07. This fact make sense, since the practical activities have to be completed individually. They can interact through the virtual platform, and with lecturers. Additionally, students think the ease of access to the ViRe-Lab is not so good with a value of 3.58 points, when comparing it with other factors, as shown in Table 2. The standard deviation for all of them are from 0.42 to 0.80, so they are consistent.

As detailed in Table 3, the 95% of the students consider ViRe-Lab to be useful or strongly useful for their experience with it. In addition to this, all of the students perceive that ViRe-Lab is easy of use or strongly easy of use, which helps to learn within the practical activity. The students’ attitude is also high, since the 95% of students agree or strongly agree with it, and there is a more or less positive collaborative factor that influences students’ mind as for the lived experience. However, the students’ perception for the easy access to ViRe-Lab is lower, although none of the students disagrees or strongly disagrees in this point. Finally, the 90% of the students were willing to use the proposed laboratory in the cloud in other Engineering activities. The rest of them were neutral about this statement.

### 5.4. Students’ Acceptance with SEM Models

A confirmatory analysis is performed to analyze the initial hypotheses defined above for the defined SEM model. Figure 12a represents the suitability the proposed hypotheses (H1 …H5) for this confirmatory analysis. Each arrow (or defined hypothesis) is labeled with a reliability influence value. Influence values over 0.4 are considered to be very reliable [43,46].

Figure 12b show the influence among factors for the proposed SEM model for the ViRe-Lab laboratory. The students’ perceived usefulness influence their attitude (H1 = 0.53), and the future intention of use this technology in indirectly (H3 = 0.72). In contrast, the students’ estimated effort to use the platform does not affect their attitude (H2 = 0.04) to use the laboratory. Neither the students’ social influence (H4 = −0.01) nor the students’ ease of access the laboratory (H4 = −0.04) affect the intention to use it for other subjects. What is more, the influence is even negative. For this reason, H2, H4 and H5 hypotheses will be discarded in an improved SEM model, which will be defined and analyzed in the rest of the Section.

As a second part of this analysis, focusing us on the initial SEM model for the laboratory presented in this work (and the conclusions obtained in the exploratory phase), an improved SEM model more suitable for this technology is proposed. H2, H4 and H5 hypotheses are discarded, as explained above, and a new two hypotheses are included in the model, H6 and H7.

**Hypothesis 6** **(H6).**
*The perceived usefulness of the laboratory will positively influence the intention of use it in the future.*


**Hypothesis 7** **(H7).**
*The ease of access to the laboratory will positively influence the the attitude of students to perform the practical activities.*


The improved SEM model is presented in Figure 13a. It can be observed that the students’ perceived usefulness of the ViRe-Lab laboratory clearly influences their attitude with the laboratory (H1 = 0.45). In addition, the students’ intention of use is positively very affected by the students’ attitude (H3 = 0.83). On the other hand, the easy access to ViRe-Lab slightly affects their attitude (H7 = 0.26). The intention to use ViRe-Lab in other contexts is also affected by the perceived usefulness (H6 = 0.41).

Finally, the most relevant statistical indexes detailed in the literature are calculated to prove the real reliability of our confirmatory analysis for the improved SEM model [4,46,47]. First, the relation Chi-square (X2) and Degree Freedom (DF) is 0.3, which is lower than 3.0, and the X2 with a value of 0.6 is higher than 0.5. Goodness of Fit Index (GFI) and Comparative Fit Index (CFI) values are 0.985, respectively. Since these values are higher than 0.9, they are very good. The Root Mean Square Error of Approximation (RMSEA) fits the model completely with a near value of 0. As a consequence, the improved SEM model and its hypotheses are tested in a satisfactory way, since the statistical values are within the expected values already studied in the literature.

## 6. Conclusions and Future Works

Nowadays, there is a huge increment for the employment of new technologies, since the way of and living working nowadays has become digital. The education context is not an exception in this sense; even more, in practical topics for the field of Engineering. This fact is more noticeable when a distance methodology is used during the learning/teaching process of students. UNED is the biggest University in Spain, and it includes a distance methodology in virtual courses. In many subjects, there are also periodically physical and/or virtual classes.

In this digital era, new threats and vulnerabilities appear, so cybersecurity competences and abilities have to be acquired by students. In this sense, this paper has shown the fundamentals of our emulated virtual remote laboratory in the cloud (named, ViRe-Lab), which has been used for a cybersecurity subject, CIS, belonging to Engineering studies. Technical requirements about the implementation, development and integration of this laboratory are detailed in the paper; in addition to how it has been integrated in the students’ learning/teaching process. The ViRe-Lab can be considered as a another integrated resource within the virtual platform.

On the other hand, the students’ interactions with the ViRe-Lab had also been previously analyzed to detect their behaviour by concluding they interact with the laboratory in a suitable way. In addition to this, a set of perceived factors (usefulness, estimated effort, social influence, attitude, ease of access intention of use) which may affect the process of students’ learning/teaching are studied from the point of view of their satisfaction with the ViRe-Lab. The satisfaction values are outstanding since the mean values of these factors are mostly higher than four points out of five. Additionally, these factors are analyzed for students’ acceptance in case of using ViRe-Lab in other Engineering subjects.

Furthermore, two structural equation models, based on TAM/UTAUT, have statistically been hypothesized and validated. From the confirmatory analysis of the improved second, it can be concluded that the students’ perceived usefulness influences their positive attitude toward the ViRe-Lab. The students’ attitudes affect their intention of using the ViRe-Lab in a very strong way, and their perceived usefulness also influences the intention of using the ViRe-Lab immediately. The students’ ease of access slightly influences their attitude. Additionally, the calculated statistical values for the improved SEM are within the expected ranges of reliability, with the X2 parameters being equal to 0.6, the relation among X2 and DF equal to 0.3, both GFI and CIF values equal to 0.985, and RMSEA very near zero.

As a future direction, we are working towards an extended approach to developing physical device remote solutions. These approaches will have advantages of the real world, i.e., the remote laboratory and all the advantages of efficient virtualization presented in this work. Therefore, there will be a laboratory of real things and micro-services, where dynamic resource provisioning and fault-tolerance features are considered. Even serious games can be employed as a complementary way.

It will be essential to define specific learning processes and monitor them to undertake projects based on IoT infrastructures in the context of learning analytics. This way, students can achieve a set of skills ranging from devices and sensors, their communication IoT protocols, the storage management, and the processing environments in the cloud for the data generated by sensors. Students must use components and layers (hardware/software) that are deployed in these types of solutions, so the learning process must incorporate the use of technological tools similar to those that will be found in these IoT environments and domains. Our research efforts will be to develop, deploy, and evaluate this technology related to cybersecurity, cloud computing, and IoT infrastructures and services topics.

## Figures and Tables

**Figure 1 sensors-20-03011-f001:**
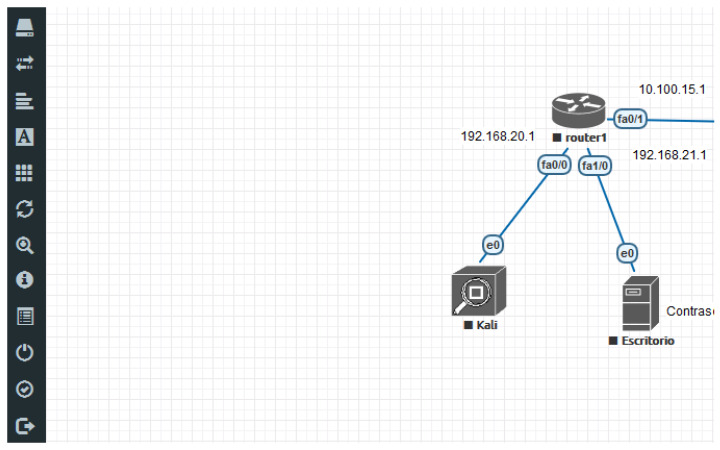
Example of Laboratory in the ViRe-Lab.

**Figure 2 sensors-20-03011-f002:**
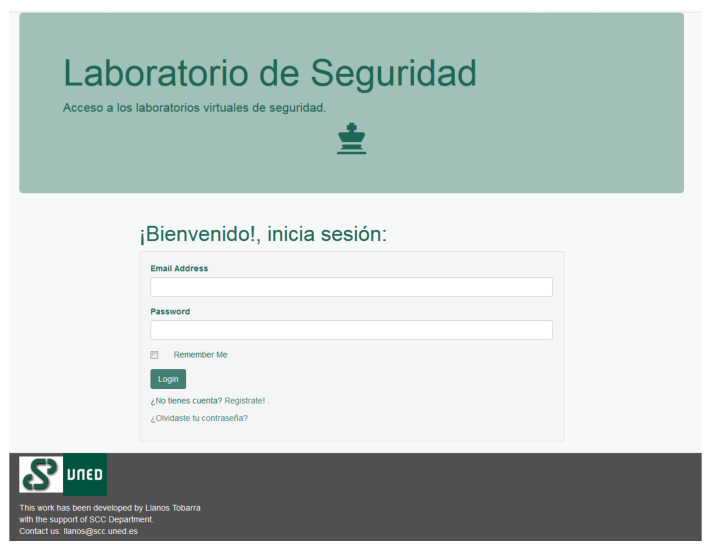
Virtual remote laboratory, ViRe-Lab.

**Figure 3 sensors-20-03011-f003:**
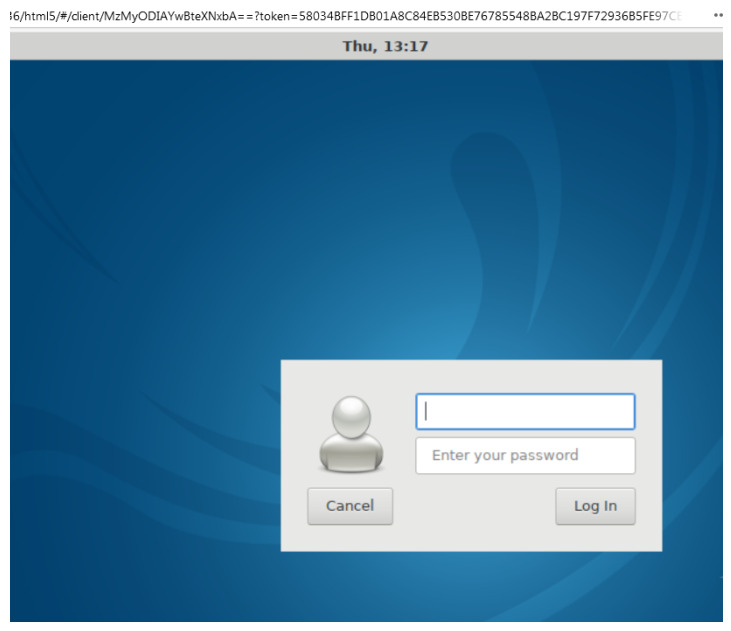
Accessing to a Network Node of the Laboratory for the CIS Subject.

**Figure 4 sensors-20-03011-f004:**
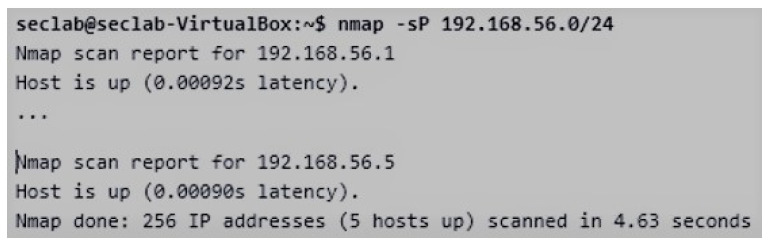
Location of Network Devices with the NMAP Tool.

**Figure 5 sensors-20-03011-f005:**
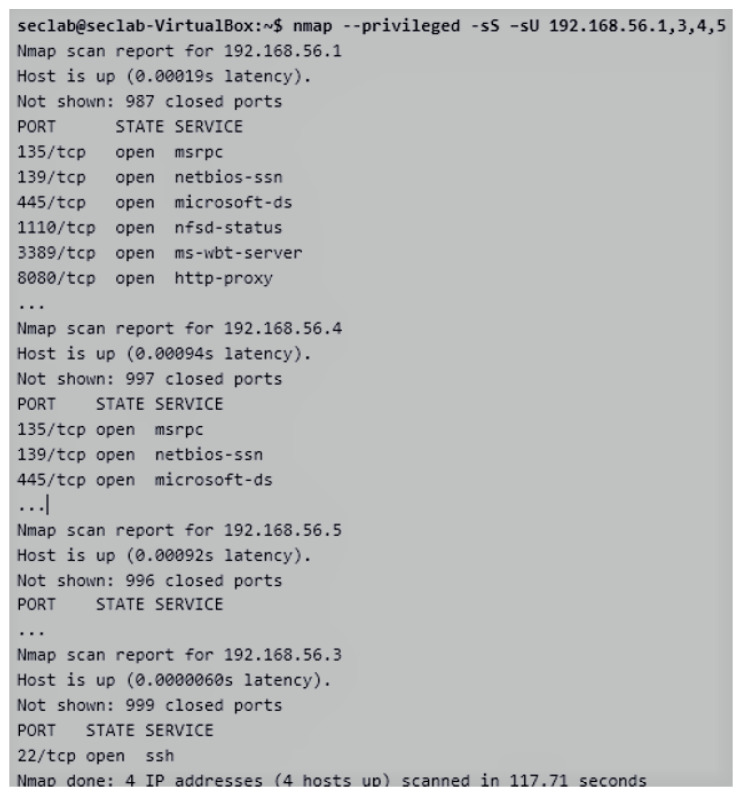
Discovering of Open Ports and Services for a set of Network Device.

**Figure 6 sensors-20-03011-f006:**
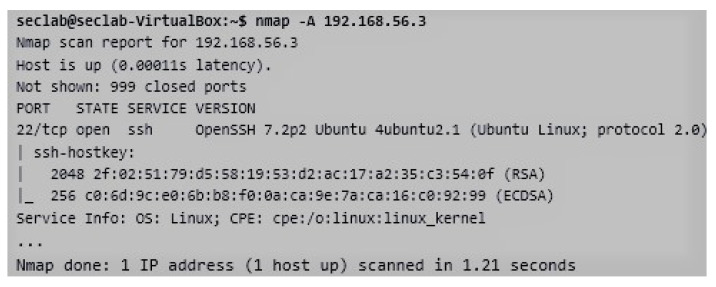
Obtaining of Port Services and Additional Information for a Network Device.

**Figure 7 sensors-20-03011-f007:**
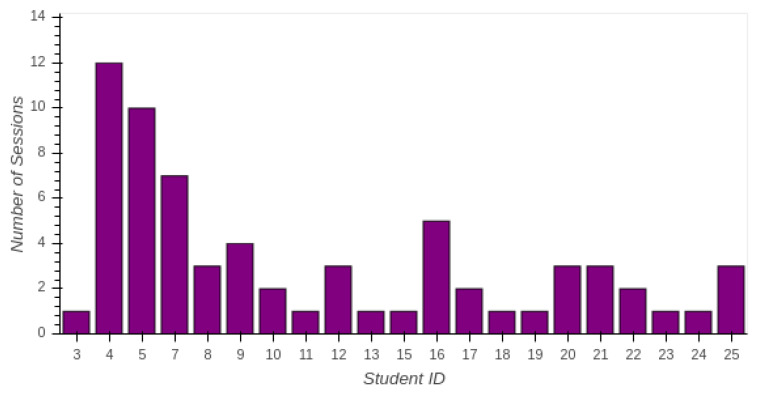
Amount of Sessions per Student.

**Figure 8 sensors-20-03011-f008:**
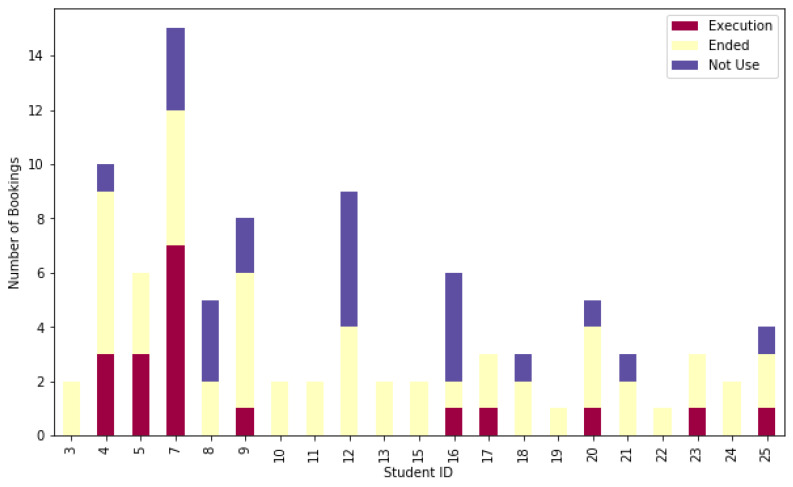
Amount of Booking Sessions per Student (Divided by Status).

**Figure 9 sensors-20-03011-f009:**
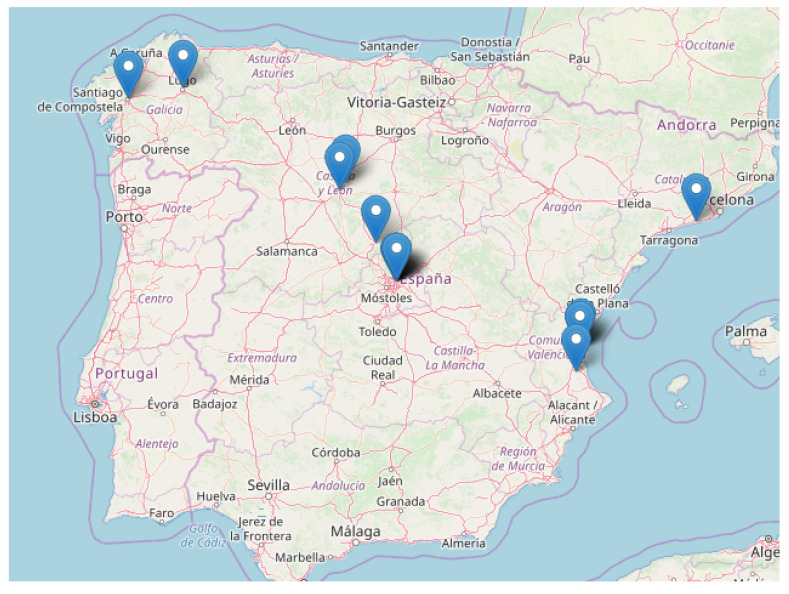
Geographical Distribution of Students.

**Figure 10 sensors-20-03011-f010:**
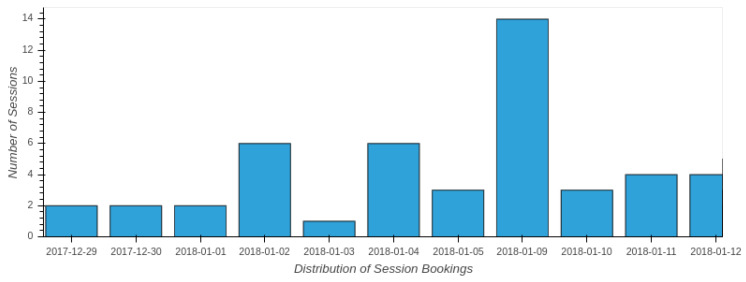
Booking Distribution (from 29th December to 12th January).

**Figure 11 sensors-20-03011-f011:**
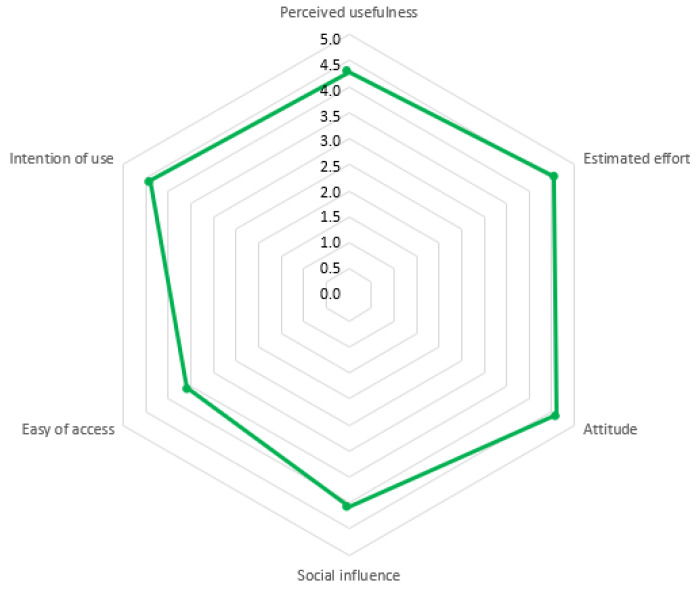
Evaluated Factors.

**Figure 12 sensors-20-03011-f012:**
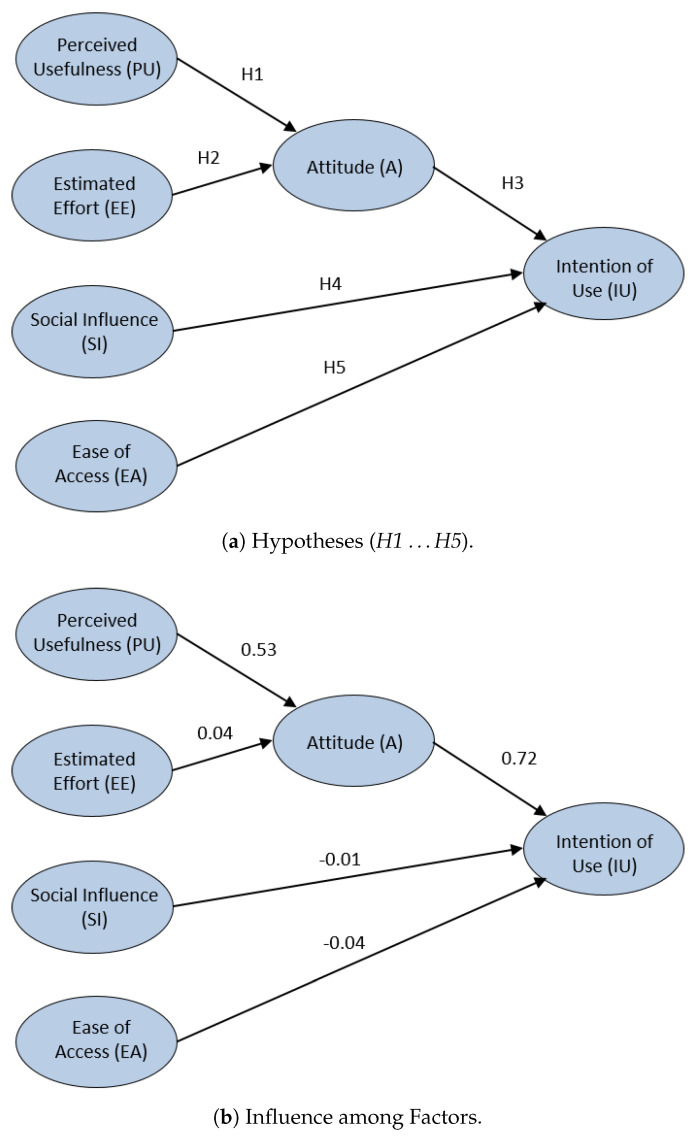
SEM for the ViRe-Lab with the UTAUT Model.

**Figure 13 sensors-20-03011-f013:**
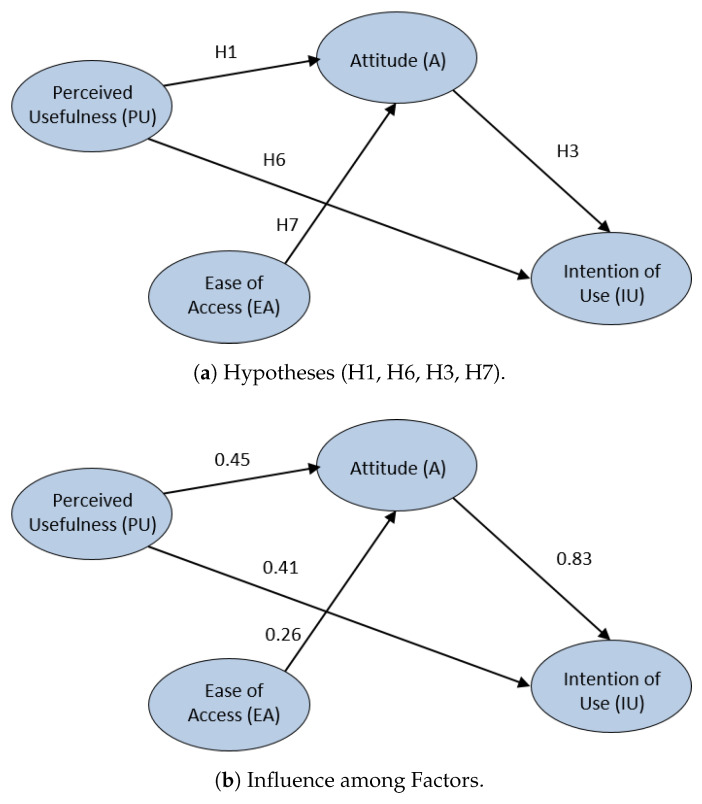
Proposed Improved UTAUT Model for the ViRe-Lab.

**Table 1 sensors-20-03011-t001:** Demographic Data.

Type	Option	Percentage (%)
Gender	Male	85%
	Female	15%
Age Group	≤30 years	40%
	30–39 years	40%
	40–50 years	15%
	≥50 years	5%
Occupation	Non-computer science related job position	5%
	Computer science related job position	80%
	Other situation	15%

**Table 2 sensors-20-03011-t002:** Mean (M)/Standard Deviation (SD) Values for each Selected Factor.

Factor	M	SD
Perceived usefulness	4.30	0.59
Estimated effort	4.55	0.42
Attitude	4.61	0.46
Social influence	4.07	0.80
Ease of access	3.58	0.65
Intention of use	4.40	0.71

**Table 3 sensors-20-03011-t003:** Results of the Survey.

Factor	Strongly Agree	Agree	Neutral	Disagree	Strongly Disagree
Perceived usefulness	45%	50%	5%	0%	0%
Estimated Effort (Ease to Use)	55%	45%	0%	0%	0%
Attitude	75%	20%	5%	0%	0%
Social influence	30%	40%	30%	0%	0%
Ease of access	10%	55%	35%	0%	0%
Intention of use	55%	35%	10%	0%	0%

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
