# Peer review of "Emulating and Evaluating Virtual Remote Laboratories for Cybersecurity†"

_sensors, 2020, doi:10.3390/s20113011_

Round 1
Reviewer 1 Report
The authors claimed to propose a virtual remote laboratory in the cloud, which has been integrated into a cybersecurity subject. The work is somehow new but several limitations from the English to the technical hinder to grasp the main point of the work, some of them are listed below: +In my opinion, the abstract is too cumbersome, and it is hard to catch the key point. The keywords need to be more detailed. + There is a need to highlight the numerical result found in this research in the abstract. +The paper structure is hard to follow and must be elaborated in the technology they applied as well as support more rigorous technical aspects. Even though, it is essential to address their method using the algorithm which makes it clear to grasp the steps of the improvements of the method. +The time and space complexity and algorithm not specified. +Test Setup and tuning for the work is expected to elaborate and detailed for future productions. +The literature has to be strongly updated with some relevant and recent papers focused on the fields dealt with the manuscript. Securing IoT-Based RFID Systems: A Robust Authentication Protocol Using Symmetric Cryptography." Sensors 19, no. 21 (2019): 4752.Author Response
Comment:
The authors claimed to propose a virtual remote laboratory in the cloud, which has been integrated into a cybersecurity subject. The work is somehow new but several limitations from the English to the technical hinder to grasp the main point of the work, some of them are listed below:
Answer:
Thank you very much for these comments. They have been very useful for improving the current version of the manuscript, as detailed next.
Comment:
+In my opinion, the abstract is too cumbersome, and it is hard to catch the key point.
Answer:
We agree with this comment. For this reason, we have re-written the abstract of the manuscript according to the provided suggestions and being consistent with the rest of the manuscript. Changes are marked in red in the manuscript for clarity.
Comment:
The keywords need to be more detailed.
Answer:
We agree with this comment. Keyworks have been changes by a set of more detailed ones.
Comment:
+ There is a need to highlight the numerical result found in this research in the abstract.
Answer:
The abstract has been re-written, re-organized, and numerical results have been included, as requested.
Comment:
+The paper structure is hard to follow and must be elaborated in the technology they applied as well as support more rigorous technical aspects. Even though, it is essential to address their method using the algorithm which makes it clear to grasp the steps of the improvements of the method.
Answer:
We agree with these comments. For this reason, we have used the reviewer’s comments to improve the different sections to make the manuscript more elaborated in the employed technology, even, including a new section for the creation of EVE-NG environments and the algorithm employed by the laboratory to control its access and the booking sessions. All improvements are marked in red color in the new version of the manuscript.
Comment:
+The time and space complexity and algorithm not specified.
Answer:
We have included the pseudocode of the algorithm employed by the laboratory management to control its access and the booking sessions. This way, the reader can check complexity of the algorithm built over the EVE-NG core.
Comment:
+Test Setup and tuning for the work is expected to elaborate and detailed for future productions.
Answer:
Technical details about the employed and tuned technology is presented in the new version of the manuscript.
Comment:
+The literature has to be strongly updated with some relevant and recent papers focused on the fields dealt with the manuscript. Securing IoT-Based RFID Systems: A Robust Authentication Protocol Using Symmetric Cryptography." Sensors 19, no. 21 (2019): 4752. DOI: 10.1049/iet-com.2019.0554 doi:10.32604/cmc.2019.06288
Answer:
We agree with this point. In order to minimize the issue, it we have extended clarifications and explanations about the state of the art and enforced this study with additional works from the literature (the three ones recommended in this comments, and other additional ones found in the literature). All new explanations are marked in red color in the new version of the manuscript. These references are from [2] to [3] (Section 1), from [20] to [24] (Section 2).
Reviewer 2 Report
The paper idea of Virtual Remote Laboratories is very interesting in the current situation the world live in. The paper needs some improvement though, as follow:
- When the authors say in the abstract "The proposed laboratory obtains very high acceptance and 11 satisfaction", what kind of satisfaction you mean? you need to better indicate that in the abstract.
- The introduction section starts well but ends vaguely. What are the contributions of this paper? add a list to the intro section.
- The methodology seems a bit superficial, with no formal modelling especially for this kind of systems. Please add some sort of modelling to the paper.
- I do not seem to understand the context of the evaluation; the authors need to re-structure and re-write this section to better indicate the overall purpose of the evaluation.
- Proofread the paper to fix all typos and grammatical errors.
Author Response
Comment:
The paper idea of Virtual Remote Laboratories is very interesting in the current situation the world live in. The paper needs some improvement though, as follow:
Answer:
Thank you very much for these comments. They have been very useful for improving the new version of the manuscript, as detailed next.
Comment:
When the authors say in the abstract "The proposed laboratory obtains very high acceptance and 11 satisfaction", what kind of satisfaction you mean? you need to better indicate that in the abstract.
Answer:
We agree with this comment. The abstract has been re-written in order to clarify the principal contributions of this manuscript. Specifically, we have clarified what we mean when we say satisfaction and acceptance. The introduction and evaluation sections have also been clarified in this aspect: some the obtained results answer the students’ satisfaction for each factor/indicators; the SEM models (and associated hypotheses) and other statistical parameters answer the general students’ acceptance for intention of use the technology in other contexts.
Comment:
The introduction section starts well but ends vaguely. What are the contributions of this paper? add a list to the intro section.
Answer:
We agree with this comment. For this reason, a new paragraph detailing the main contributions of the paper is included in the new version of the manuscript. Additionally, the introduction section has been revised, and the transition paragraph between section 1 and 2 has been re-written. Changes are marked in red in the manuscript for clarity.
Comment:
The methodology seems a bit superficial, with no formal modelling especially for this kind of systems. Please add some sort of modelling to the paper.
Answer:
We agree with this comment. The methodology of the new version of the paper has been revised, as well as more formal description and modeling of the technology is included in the new version of the manuscript. All improvements included in the new version of it have been marked in red color.
Comment:
I do not seem to understand the context of the evaluation; the authors need to re-structure and re-write this section to better indicate the overall purpose of the evaluation.
Answer:
We agree with this comment. For this reason, we have revised this section for clarification purposes as for the overall purposes of the evaluation, in terms of interactions, satisfaction, and acceptance. Some of the titles have been varied for this purpose. Changes are marked in red in the manuscript for clarity.
Comment:
Proofread the paper to fix all typos and grammatical errors.
Answer:
We agree with this comment. The language employed along the manuscript has been revised in order to improve the English edition of the manuscript.
Reviewer 3 Report
The authors present a very interesting paper on the inclusion of cybersecurity in the curricula of engineering students to make it part of their professional skills.
I think that the work is appealing for a wide group of the scientific community, especially in the current times where online education has so much presence, accentuated even more by the covid-19 pandemic.
In this regard, I would like to offer some suggestions(included in the pdf attached) that could improve the clarity of certain aspects of the paper.

Author Response
Comment:
The authors present a very interesting paper on the inclusion of cybersecurity in the curricula of engineering students to make it part of their professional skills.
I think that the work is appealing for a wide group of the scientific community, especially in the current times where online education has so much presence, accentuated even more by the covid-19 pandemic.
In this regard, I would like to offer some suggestions (included in the pdf attached) that could improve the clarity of certain aspects of the paper.
Answer:
Thank you very much for these comments. We have revised and incorporated them to the new version of the manuscript. The specific reviewer’s comments are detailed below. There are other marked texts without a comment, they have been considered too for the new version of the manuscript.
Comment:
(ABSTRACT) This point is interesting, since it is online and it is decided to present it in the cloud my only concern is, what is different and makes a significant contribution to what is already working?
Answer:
We agree with this comment. For this reason, we have re-written the abstract of the manuscript according to the provided suggestions, detailing the principal contributions, and being consistent with the rest of the manuscript. Changes are marked in red in the manuscript for clarity.
Comment:
(ABSTRACT) At some point in the paper the authors also mention the KPIs (page 4) but I suggest that the detail of this indicator collection be further clarified (previous studies?).
Answer:
We agree with this comment. This point can confuse the reader, since KPIs and indicators are not the same ones. Indicators refer to the physiological factors to be used in the evaluation section, whereas KPIs are performance indicators and metrics to stablish a security policy in the curricula of the subjects. For this reason, the “indicator” word has been renamed as “factor”. These meanings are clarified in the new version of the manuscript and marked in red for clarity.
Comment:
(1. Introduction) Although it is somehow mentioned implicitly, what I really miss is that it clarifies more thoroughly the objective of the paper, apart from talking about the virtual laboratory or saying that this is the contribution. This point needs a little strengthening.
Answer:
We agree with this comment. For this reason, we have re-written the abstract, included a new paragraph in section 1 detailing the main contributions of this work, among other related improvements for the new version of the manuscript.
Comment:
(3.1. Educational Contextualization) Please, clarify how the authors choose these KPI.
Answer:
As detailed above, this aspect has been clarified in the new version of the manuscript,
Comment:
(4.1. The Purpose and Access to the Laboratory) This point has to be reinforced in the introduction.
Answer:
We agree with this comment. As stated above, some efforts about this point have been performed in the abstract and introduction of the new version of the manuscript.
Comment:
(SEM Models). Please, improve the quality of these graphics.
Answer:
We have tried to improve the quality of these graphics. If needed, we can provide the sources of these graphics.
Comment:
(6. Conclusions and Future Works) the future line is somewhat weak, there is a lot of work and research done on cybersecurity, so what does this proposal offer as a substantial difference from what is working right now?.
Answer:
We agree with this comment. The conclusions and future works section has been revised and extended in this aspect to enforce the further research. The introduction and state of the art sections have also revised for this purpose.
Round 2
Reviewer 2 Report
All my comments have been addressed, the paper is ready for publication in the present form
This manuscript is a resubmission of an earlier submission. The following is a list of the peer review reports and author responses from that submission.